# REVERSE CHAIN: A GENERIC-RULE FOR LLMs TO MASTER MULTI-API PLANNING

## ABSTRACT

While enabling large language models to implement function calling (known as APIs) can greatly enhance the performance of LLMs, function calling is still a challenging task due to the complicated relations between different APIs, especially in a context-learning setting without fine-tuning. This paper proposes a simple yet controllable target-driven approach called Reverse Chain to empower LLMs with capabilities to use external APIs with only prompts. Given that most open-source LLMs have limited tool-use or tool-plan capabilities, LLMs in Reverse Chain are only employed to implement simple tasks, e.g., API selection and argument completion, and a generic rule is employed to implement a controllable multiple functions calling. In this generic rule, after selecting a final API to handle a given task via LLMs, we first ask LLMs to fill the required arguments from user query and context. Some missing arguments could be further completed by letting LLMs select another API based on API description before asking user. This process continues until a given task is completed. Extensive numerical experiments indicate an impressive capability of Reverse Chain on implementing multiple function calling. Interestingly enough, the experiments also reveal that tool-use capabilities of the existing LLMs, e.g., ChatGPT, can be greatly improved via Reverse Chain.

## 1 INTRODUCTION

Recently, there has been an impressive wave in the progress made in Large Language Models (LLMs), allowing them to demonstrate excellent performance in a variety of tasks, such as powerful conversation, in-context learning, and code generation (Chowdhery et al., 2022; Brown et al., 2020; Scao et al., 2022; Wei et al., 2022a). However, LLMs still face difficulties with some specialized tasks because they are fundamentally limited by the information they stored and learned. The learned information can become outdated and may not be suitable for all applications. A simple way to overcome these limitations is to integrate LLMs with external tools (known as APIs) so that LLMs can access up-to-date knowledge, run computations or use other external services. In this scenario, LLMs can be regarded as a controller which not only needs to understand user intent but more importantly, they are required to be able to select the appropriate tools and orchestrating them to accomplish the given task.

Although LLMs demonstrate excellent language comprehension and reasoning abilities, unfortunately they still lack the sophistication to fully understand human instructions and effectively implement function calling. In order to augment the ability of LLMs on function calling, the existing solutions can be categorized into two main approaches: fine-tuning and in-context learning. In the fine-tuning approach, instruction tuning datasets for function calling are needed which often requires extensive human annotations or demonstrations. Additionally, the fine-tuned LLMs is likely to be restricted to a predefined external tools and might be hard to adapt to the newly registered APIs Liang et al. (2023); Qin et al. (2023b); Li et al. (2023); Schick et al. (2023). Unlike the expensive fine-tuning approach, in-context learning paradigm only needs to provide the instructions with a few shots of demonstrations, which allows developers easily to register new APIs to the LLM platform. In this work, we mainly focus on augmenting the existing capabilities of LLMs on function calling in an in-context learning setting.

| Task Type | Example | API planning |
|---|---|---|
| Single-tool | What's the weather in New York ? | getWearther(city='New York') |
| Independent multi-tool | What's the weather in New York? When's my next meeting? | getWearther(city='New York') showCalendar(event='next meeting') |
| Composition multi-tool | I have headache, can you help me make an appointment? | makeAppointment ( hospital= getHospital(symptom='headache')) |

Table 1: Different types of API planning based on the number of tools and tool dependency.

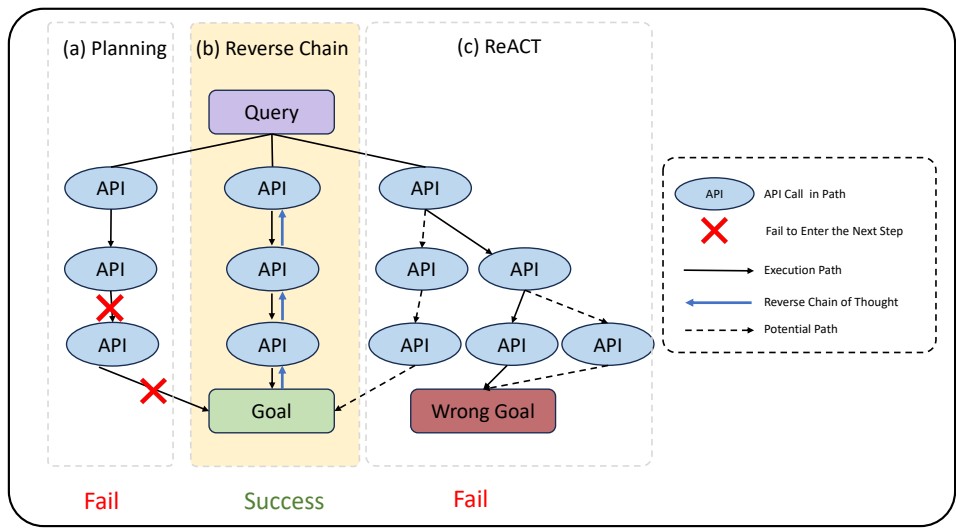

Figure 1: A comparison of our Reverse Chain with one-step function Planning and ReAct for multi-API planning.

The emergence of multiple external APIs adds complexity to the task of function calling for LLMs, especially for in-context learning. Different from the existing work (Li et al., 2023; Qin et al., 2023b; Tang et al., 2023) which mainly focuses on single-tool tasks or independent multi-tool tasks as shown in Table 1, this paper targets at empowering LLMs to be able to handle a much more complicated type of tasks (called composition task) where multiple possibly dependent APIs are needed. The single-tool task and independent multi-tool tasks can be regarded as special cases of composition multi-tool task, thus the proposed approach has the ability to handle them as well with minor change.

To solve composition multi-tool task, one straightforward solution is to let LLMs directly output a multi-function planning in one step as shown in Figure 1. Apparently, the accuracy of the **one-step planning** algorithm tends to be low, particularly in complex scenarios that involve ambiguity (Shen et al., 2023). In order to alleviate these issues, CoT (Chain of Thought) (Wei et al., 2022b) proposes a **step-by-step intermediate reasoning process** that involves decomposing tasks into individual and simpler sub-tasks. This decomposition enhances the reasoning abilities, ultimately leading to a better accuracy to some extent. In addition to taking advantage of reasoning, a more advanced approach called ReAct(Yao et al., 2022) also employs actions to gather additional information to improve reasoning performance. However, as shown in Figure 1, in a multi-function calling scenario, ReAct might not necessarily follow the right reasoning path to achieve goal. Our experiments indicate hard-to-control issues with the CoT and ReAct approaches: each step of these methods exhibit a high level of **unpredictability** and **uncertainty**, where errors might propagate due to a wrong thought or action, leading to an incorrect solution path.

In order to alleviate these issues, we propose a controllable yet general approach called **Reverse Chain**. Different from CoT which requires LLMs to have certain level of reasoning abilities, the proposed Reverse Chain is a combination of a generic rule and LLMs where LLMs are employed to implement API selection and argument completion based on API description and a generic rule is to

decompose multi-API planning into API selection and argument completion such that the strengths and capabilities of LLMs can be fully leveraged without expecting LLMs to have strong reasoning abilities. Specifically, Reverse Chain performs a multi-API planning task in a reverse manner: Starting from selecting the final API of a given task, each preceding step is inferred backward. Additional rule-based constraints are employed to limit the order of planning: first API selection, then argument completion from query and context, and finally backward inference to figure out which API's output can properly complete the missing arguments before asking user that information. This process continues until a given task is completed. In summary, the contributions of this paper are

1. A target-driven approach called Reverse Chain is proposed to enhance the API planning capability of LLMs in an in-context learning setting. This approach utilizes step-by-step problem-solving solution, making it manageable for LLM to handle. Additionally, by employing a generic rule to limit the specific order of function planning, the process becomes more controllable, which is suitable to deploy in production.

2. Extensive experiments are conducted to demonstrate the superiority of the Reverse Chain approach over the state-of-the-art in-context learning approaches, e.g., CoT and ReAct. Furthermore, we have also shown the effectiveness of applying the Reverse Chain technique to empower the ability of LLMs on multi-API calling.

## 2 OUR MULTI-API PLANNING APPROACH

In order to address the limitations of the CoT and ReAct method, here we introduce a controllable yet general approach called **Reverse Chain**. Different from CoT and ReAct, a step-by-step problem-solving path in Reverse Chain is predefined using a generic rule. However, this generic rule is not restricted with a certain type of tasks. Interestingly Reverse Chain performs a task decomposition in a reverse manner: Starting from selecting the final API for a given task, following by argument completion, then selecting a proper API again to complete the missing argument. This process is repeated until all arguments in the involved APIs are completed. Note that driven by the target, each preceding step in Reverse Chain is inferred backward, which makes it less prone to deviate from the intend path, unlike ReAct shown in Figure 1. Given conversation context, user query and a set of APIs, the generic rule on specifying the multi-function planning order in Reverse Chain is outlined as follows:

1. The first step is to employ LLMs to select a proper API which can directly handle a task of interest. This step is referred to as **API Selection**.

2. After finding the required arguments of the API, the second step is to utilize LLMs to implement argument completion. This step is referred to as **Argument Completion**.

Note that if LLMs could not complete all the required arguments at Step 2, instead of directly asking user for the value of the missing argument, the generic rule will first pre-select some APIs with whose output matches the type of the required arguments and then redefine selecting a proper API to complete the missing argument as the current task of interest. The generic rule let LLMs go back to Step 1 to continue the process. It is worth mentioning that Step 1 and Step 2 are executed iteratively until the termination condition is met, e.g., none of APIs could be selected to complete the missing arguments or all of the required arguments are completed. For the first case, the generic rule will take an extra step to ask user that missing information directly such that all involved APIs are ready to be executed. In the following we will use a specific example to introduce the workflow of Reverse Chain to better interpret its concept.

### 2.1 WORKFLOW OF REVERSE CHAIN

Assume our current task is to satisfy user query "Please help Jack book a meeting room from 9:00am to 10:00am.", and we have three APIs in our API store with their descriptions, arguments, and output shown in Table 2. Note that the ground truth API planning is shown as follows:

$$\textbf{BookRoom}(\text{person\_ID} = \textbf{Name2ID}(\text{person\_name='Jack'}),$$
$$\text{room\_ID} = \textbf{RecommendRoom}(\text{start\_time='9am'}, \text{end\_time='10am'}),$$
$$\text{start\_time} = \text{'9am'}, \text{end\_time='10am'})$$

The workflow of Reverse Chain is provided as follows:

| API | Description | Arguments | Output |
|---|---|---|---|
| Name2ID | Convert user name to user ID | person_name | person_ID |
| RecommendRoom | Recommend the ID of an available meeting room | start_time
end_time | room_ID |
| BookRoom | Book a meeting room | person_ID
room_ID
start_time
end_time | room_Info |

Table 2: The definition of APIs to handle a given user query "Please help Jack book a meeting room from 9am to 10am".

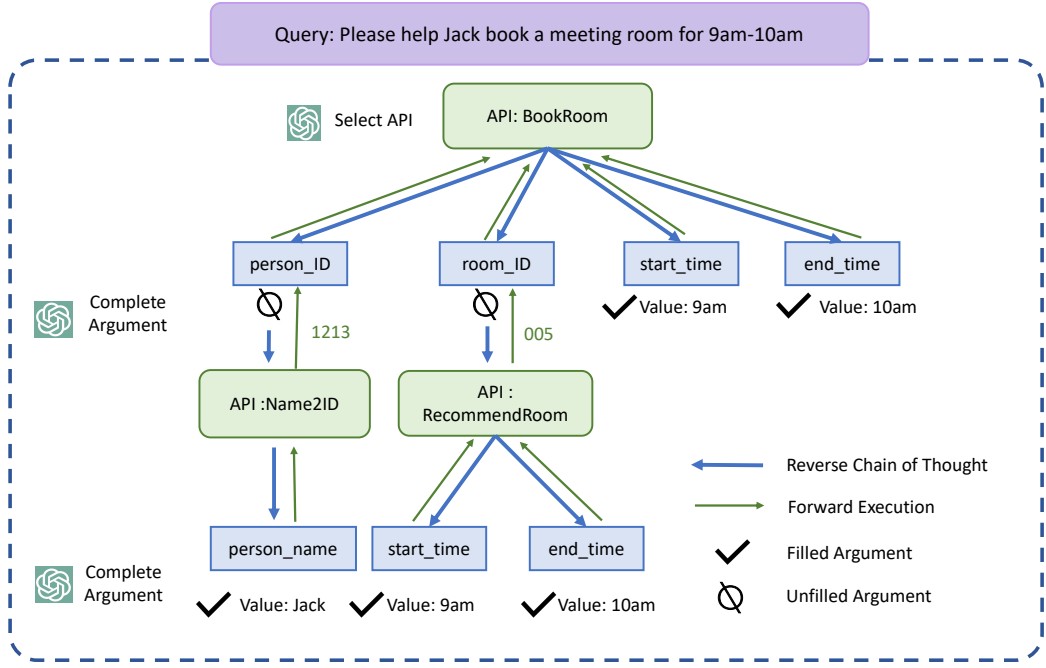

Figure 2: Workflow of Reverse Chain to satisfy a specific user query

1. **API Selection.** Firstly, LLM identifies the ultimate API based on conversation context, user query and API description, and then select a matching API. In this example, LLM select an API named **BookRoom** to match "booking a meeting room". The prompt used in this step is illustrated in Figure 3.(a).

2. **Argument Completion.** Before implementing argument completion, the required arguments of the API are identified through engineering guidance. In this example, **BookRoom** has four required arguments, that is, person_ID, room_ID, start_time, and end_time. Based on the query and the description of the candidate APIs, LLMs is used to complete these four arguments at the current step. There are three possible outcomes:
Case 1. The argument value extracted directly from the context and user query.
Case 2. Another API name with whose output could complete the missing argument, indicating LLMs could not obtain the argument value directly. Note that the arguments of this new API has to been completed before function calling execution.
Case 3. None, representing the inability to obtain the argument value from context, user query and the possible API output. In this case, the generic rule will request the argument value directly from the user.

In this example, plugging in Reverse Chain with ChatGPT leads to the following results:

Extract the argument person_ID − − − − > API: **Name2ID**.

Extract the argument room_ID − − − − > API: **RecommendRoom**.

Extract the argument start_time − − − − > Value: 9:00am.

Extract the argument end_time − − − − > Value: 10:00am.

Thus, the completed arguments are start_time = "9:00am" and end_time = "10:00am", while the required arguments for person_ID and room_ID remain unfilled since the selected APIs **Name2ID** and **RecommendRoom** still have missing arguments.

The specific prompt used in this step is illustrated in Figure 3.(b). It is worth noting that some prompt optimizations work might be needed at this step, and we further investigate different argument completion approaches in the ablation study of Section 4.3.

3. *Argument completion for internal selected APIs* Although LLMs have figured out the right internal APIs to complete missing arguments at Step 2, in order to execute these APIs, the required arguments of these internal APIs still needs to be completed at the current step. Specifically, the process starts with completing argument person_ID and then proceeds to argument room_ID where each argument is completed in a similar way as introduced at Step 2. The generic rule first queries the required argument of API Name2ID, which is person_ID, and then let LLMs extract possible values to complete this argument:

Extract the argument person_name − − − − > Value: Jack.

Note that now all required arguments of **Name2ID** are filled successfully, and the API **Name2ID** is ready to be executed. As a result, the arguments of person_ID in API **Book-Room** is filled successfully. Since the values of the required arguments of **Recommend-Room** are already found at Step 2, **RecommendRoom** is now ready to be executed as well. As a result, the arguments of room_ID in API **BookRoom** is filled successfully.

4. **Multi-API Execution.** All of the required arguments for the related APIs **BookRoom**, **RecommendRoom** and **person ID** have been completed. Execution can be started by a bot engine.

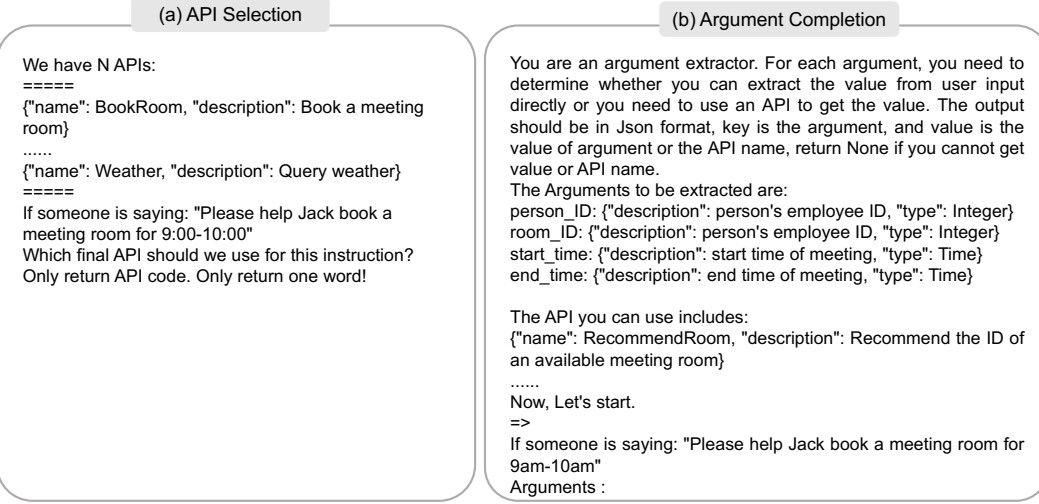

Figure 3: The details of prompts used in Reverse Chain for API Selection and Argument Completion.

# 3 EXPERIMENTS

In this section, extensive experiments are conducted to investigate the performance of Reverse Chain. We start with generating an evaluation dataset, benchmarking different LLM methods on function

calling and defining the evaluation metrics. In Section3.1, to benchmark Reverse Chain, we compare its API planning capabilities with the current state-of-the-art in-context learning solutions. In Section3.2, a series of ablation experiments are conducted to illustrate the rationale behind Reverse Chain with further analysis on the effectiveness of Reverse Chain provided in Section 3.3.

**Dataset** The dataset construction starts from selecting several APIs from the public API repository, e.g., API-Bank Li et al. (2023). Subsequently, we manually construct a set of queries associated with those selected APIs, along with their corresponding ground truth API plannings. Then, these manually constructed samples were provided as in-context examples for GPT-4 to generate more complicated new labeled samples. Furthermore, the generated samples by GPT-4 were manually verified. The prompt for dataset construction is in Appendix A.1. Note that the final dataset consists of 126 APIs and 492 labeled samples, encompassing various domain types.

In the dataset, each API is represented as a JSON object with the following fields: {name, description, arguments, output, format}. Each argument contains information about "description" and "type". One example of APIs is provided in the Appendix A.2.

According to the nesting level of API, samples in the generated dataset are categorized into three levels: level-1, which has two levels of API nesting, 352 in total; level-2, which has three levels of API nesting, 90 in total; level-3, which has four levels of API nesting, 43 in total. There are still seven samples with only one API.

**Baseline** To benchmark Reverse Chain, we compare its performance with the other five in-context learning approaches, namely **Zero-Shot**, **Few-Shot**, **Zero-Shot-CoT**, **Few-Shot-CoT**, and **ReAct** with ChatGPT being a base LLM model. In all six approaches, we incorporate API information into the prompt and rely solely on the in-context learning capability of the LLM for API planning. In the **Zero-Shot** approach, the prompt include all the information from the API candidate set and the user query. In addition to that, we add examples to the prompt in the **Few-Shot** approach. In comparison to zero-shot, the **Zero-Shot-CoT** approach includes a "think step by step" instruction in the prompt, guiding the LLM to decompose the problem into intermediate steps. Similarly, the **Few-Shot-CoT** approach augmented the few-shot approach by including explanations of the step-by-step execution of APIs in the provided examples. Lastly, we employ the langchain framework to implement the ReAct approach, which utilizes a (thought, action, observation) format template to guide the LLM in accomplishing tasks via reasoning and action. This framework enables the LLM to generate responses by following a logical sequence of thinking, taking action, and observing the results.

The examples of prompt for zero-shot and few-shot are presented in Appendix A.3. All experiments are conducted on the state-of-the-art Large language model GPT-3.5-turbo with the gpt-3.5-turbo-0301 checkpoint. The temperature is set to 0.1.

**Metrics** We use accuracy as a metric to evaluate API planning, which consists of two aspects: API name and API arguments. The argument consists of direct value filling or another API calling.

In Reverse Chain, we ensure that the output follows the aforementioned format. Consequently, character matching is employed to automatically compare the predictions with the corresponding label in terms of API name and arguments. A prediction is correct only if it matches exactly the corresponding label. Nevertheless, in other methods except Reverse Chain, the predictions may not strictly adhere to the aforementioned format, for example, they may contain thoughts,action or other details. where human evaluation is utilized to provide further assessment.

## 3.1 MAIN RESULTS

Throughtout the experiments, the given API candidates set only includes the needed APIs for a given task since the focus of this paper is primarily on evaluating the capability of LLMs on generating a proper API calling. The accuracy of various in-context learning methods is provided in Table 3. The results in Table 3 indicate that LLM exhibits limited ability on API planning in a **Zero-Shot** setting, with an accuracy around 61.0%. It is not surprising that the use of **Few-Shot** methods can enhance performance to some extent, increasing the accuracy to 88.6% compared to **Zero-Shot**. Furthermore, the incorporation of CoT has shown to be able to improve the performance of API planning with accuracy being 82.5% in **Zero-Shot-CoT**, highlighting the effectiveness of decomposing complex tasks into simple subtasks in API planning. In contrast, the **Reverse Chain**

| Method | level 1 | level 2 | level 3 | Overall |
|---|---|---|---|---|
| Zero-Shot | 66.8 | 51.1 | 28.0 | 61.0 |
| Few-Shot | 93.2 | 75.6 | 76.7 | 88.6 |
| Zero-Shot-CoT | 87.8 | 66.7 | 69.8 | 82.5 |
| Few-Shot-CoT | 93.2 | 72.2 | 48.8 | 85.1 |
| ReAct | 44.2 | 24.8 | 2.58 | 38.9 |
| Reverse Chain | **96.9** | **92.2** | **83.7** | **94.9** |

Table 3: Evaluation results on various in-context learning methods. We can observe that the proposed Reverse Chain outperforms all other approaches. T represents the temperature of GPT-3.5.

achieves the best performance. This makes sense since the proposed Reverse Chain approach not only decomposes a complicated multi-API calling problem into two types of simple tasks to let LLMs handle but also provides a target-driven nature to mitigate uncertainty. Notably, even in a zero-shot setting, **Reverse Chain** surpasses the performance of **Few-Shot-CoT** and **Few-Shot**. Although the **ReAct** method adopts reasoning and action in a step-by-step way, it relies only on the preceding observations without a global perspective, resulting in a relatively lower accuracy for multi-API planning.

Additionally, Table 3 presents the results at different levels of API planning (higher level means more difficult to planning). In Table 3, it is expected that as API planning becomes more challenging, the error rate of Reverse Chain increases. However, in more difficult scenarios, the Reverse Chain approach shows a more significant improvement (larger gap) compared to other methods, highlighting its crucial role in addressing complex multi-API calling tasks.

## 3.2 ABLATION STUDY

In this section, we mainly focus on exploring the impact of creativity of LLMs and different argument completion order on the performance of Reverse Chain.

### 3.2.1 CREATIVITY AND IMAGINATION OF LLMS ON REVERSE CHAIN

We first investigate the impact of LLM's temperature on Reverse Chain. Temperature controls the randomness of the LLM's output. A lower temperature results in more focused and deterministic responses from LLM, while a higher temperature generates more diverse and creative answers. Table 4 shows that Reverse Chain performs better at lower temperatures as we require it to make rational and accurate decisions, whereas the accuracy decreases when it seeking more creative responses.

| Method | level 1 | level 2 | level 3 | Overall |
|---|---|---|---|---|
| Temperature=0.1 | **96.9** | **92.2** | **83.7** | **94.9** |
| Temperature=0.5 | 93.2 | 90.0 | 76.7 | 91.3 |
| Temperature=1 | 84.1 | 74.4 | 76.7 | 81.9 |

Table 4: The impact of different temperatures of LLMs on the performance of Reverse Chain.

### 3.2.2 ARGUMENT COMPLETION OPTIMIZATION

| Reverse Chain | Reverse Chain_one-by-one | Reverse Chain_three-step |
|---|---|---|
| **94.9** | 72.2 | 46.9 |

Table 5: Ablation study for the design of Argument Completion in Reverse Chain.

As presented in Table 5, a series of ablation studies are performed to examine various optimizations during the development of the Reverse Chain Algorithm. As described in Section 2, the Reverse Chain approach comprises two steps: API Selection and Argument Completion. The optimizations discussed in this section primarily concentrate on enhancing the Argument Completion step.

**Reverse Chain_one-by-one** In the current Reverse Chain approach, LLM returns extraction results of all arguments at once, while an alternative approach is to have the LLM process one argument completion at a time. This parallized extraction of all arguments is referred to as Reverse Chain_one-by-one. According to the results shown in Table 5, Reverse Chain achieves an accuracy of 94.9%, outperforming Reverse Chain_one-by-one, which only achieves an accuracy of 72.2%. This difference in performance can be attributed to the fact that, unlike Reverse Chain_one-by-one, LLM in Reverse Chain has access to all information about unfilled arguments during the argument completion process. This access to comprehensive information allows for a more accurate filling of arguments. This can be illustrated through the following example, which presented in Table 6:

User query is *"help me book a flight from London to Los Angeles"*, and the API FlightBooking requires two arguments to be filled: departure_point and destination. In Reverse Chain_one-by-one, both arguments are completed with value 'London', since the LLM tend to interpret the location information in the query as the destination. However, in Reverse Chain, the LLM is aware that there exists two arguments departure_point and destination, so it will differentiate the meaning of the two locations in the query.

|  | departure_point | destination |
|---|---|---|
| Reverse Chain_one-by-one | London | **London (incorrect)** |
| Reverse Chain | London | Los Angeles |

Table 6: Examples of Reverse Chain_one-by-one and Reverse Chain

Furthermore, in addition to its superior performance, Reverse Chain is also more efficient in terms of time and computational resources since it only requires one interaction with the LLM.

**Reverse Chain_three-step** In the current Reverse Chain scheme, both query and API candidate sets are provided to the LLM for Argument Completion. The LLM can either extract values directly from the query or select a suitable API. However, in the Reverse Chain_three-step setting, argument completion is split into two steps: 1. Only the query is given for value extraction by the LLM; 2. If the LLM returns "None", it proceeds to the API selection stage, where the LLM chooses from the API candidate set. However, this method only achieved a 46.9% accuracy rate. The reason behind this is the lack of API information in the value extraction step, which often results in forcefully extracting incorrect argument values even with low certainty. By incorporating API information, the LLM can make an informed decision between utilizing the APIs or forcefully extracting the argument value, resulting in significantly improved accuracy. There is an example:

User query is *"help Jack book a meeting room"*, and one of arguments of the API BookRoom is person_ID. In Reverse Chain_three-step setting, LLM identifies 'Jack' as the value of person_ID, since person_id is a confusing concept, the answer 'Jack' is not surprising. However, in Reverse Chain, when LLM recognizes a more reliable option and discovers that the argument person_ID can be obtained using the API PersonName2ID, it disregards the extracted value 'Jack' directly.

## 3.3 WHY REVERSE CHAIN WORKS?

|  | Wrong Final Tool | Wrong Argument_API | Wrong Argument_Value | Others |
|---|---|---|---|---|
| Zero-Shot | 20 | 138 | 25 | 9 |
| Few-Shot | 17 | 19 | 16 | 4 |
| Zero-Shot-CoT | 39 | 29 | 10 | 8 |
| Few-Shot-CoT | 29 | 29 | 9 | 6 |
| ReAct | 84 | 139 | 18 | 59 |
| Reverse Chain | 14 | 3 | 8 | 0 |

Table 7: Error cause statistics all methods.

In this section, we analyze common errors in API planning individually and explain how Reverse Chain effectively addresses them for optimal outcomes. We manually go through and count the mistakes in API planning, which are categorized into three main types. The results are presented in Table 7.

The first type of mistake is called **Wrong Final Tool**. This occurs when the answer lacks the final API, resulting in termination at a wrong API and the inability to complete the instruction. As shown in Table 7, this is common in all comparison methods, as they all involve planning from scratch, which increases the likelihood of deviating from the final goal. ReAct, in particular, is more prone to this error due to its thought-action-observation approach that lacks global planning. In contrast, Reverse Chain reduces this error by starting from the final goal in a reverse manner, errors only occur when the query's final intention is unclear.

The second mistake is **Wrong Argument_API**, which occurs when the ground truth argument is the output of another API, but the predicted result does not use the API, instead fills in an incorrect value that does not meet the requirements. For example, if the argument is person_ID=PersonName2ID(name='Jack'), but the prediction is person_ID='Jack'. This type of error occurs due to potential errors in the intermediate steps during planning, thought, and action. However, in the argument completion phase of the Reverse Chain, we can avoid forcefully extracting an incorrect value for the argument by using the optimization approach in Section 3.2.2, i.e., the LLM can choose whether to use the API or extract the argument value.

The third type of error is the **Wrong Argument_Value**, which refers to incorrect extracted values for the argument. Specific cases and optimization proposals for Reverse Chain are discussed in Section 3.2.2.

## 4    RELATED WORK

**Tool Learning** The discussion of tool usage in LLMs has grown significantly, with models like Toolformer leading the way (Schick et al., 2023; Lazaridou et al., 2022; Nakano et al., 2021). Current approaches can be divided into two categories. The first category focuses on enhancing the tool-specific capabilities of compact language models through fine-tuning with specialized datasets (Patil et al., 2023; Qin et al., 2023b; Schick et al., 2023; Tang et al., 2023; Parisi et al., 2022; Qin et al., 2023a). The second category leverages the capabilities of LLMs to interact with various tools, ranging from highly specialized ones like code interpreters (Gao et al., 2023) and AI models Shen et al. (2023) to more versatile tool sets (Li et al., 2023; Liang et al., 2023). Generally, the fine-tuning method produces better results than the prompting method, however, there is still significant room for improvement in the prompting method for API planning. This work is proposed to enhance the API planning capability of prompting methods.

**Prompting LLMs** Prompting facilitates LLMs to decompose high-level tasks into sub-tasks. There are several methods to enhance prompting capabilities, such as CoT (Wei et al., 2022b) for decomposing high-level tasks, ReAct (Yao et al., 2022) for integrating reasoning and acting, and DFSDT (Qin et al., 2023b) for selecting the most promising reasoning path. All of these methods start planning or executing from scratch to the final goal, which results in high unpredictability and uncertainty. Therefore, this paper proposes a new Reverse Chain prompting approach for performing API planning, which is controllable yet feneral approcah.

## 5    CONCLUSION AND FUTURE WORK

Traditional fine-tuning LLMs with tool demonstration data can be costly and hard to generalize to new registered APIs. In this paper we have proposed a target-driven approach called Reverse Chain to empower LLMs with capabilities to use external APIs in an in-context learning setting. In Reverse Chain, a generic rule was employed to decompose a complicated multi-function calling problem into two types of simple tasks for LLMs, that is API selection and argument completion. Extensive experiments have shown that Reverse Chain could efficiently improve the tool-use capabilities of the existing LLMs, e.g., ChatGPT. Moreover, combination of a generic rule with LLMs enabled reverse-chain to attain a better performance compared to CoT and ReAct. Exploring reverse-chain with multiple different reasoning paths seems to be an interesting direction for future work.

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

# A  APPENDIX

## A.1  PROMPT FOR DATASET CONSTRUCTION

```
                          Dataset Construction Prompt

Your task is to first generate multiple APIs with their descriptions, and then generate a pair of user query and the
corresponding system response only using the predefined APIs in a nested manner, which means the output of one
API is the input of another API. Note that for each user query, system response had better employ at least four APIs.
Here is an example:
Example:
"APIs": [
        {
          "name": "GetWeatherForecast",
          "Description": "This API returns the weather forecast of a specified city on a specific date.",
          "input_params": {
            "city": {
              "description": "the name of the city",
              "type": "String"
            },
            "date": {
              "description": "the specific date",
              "type": "Date"
            }
          },
          "output_params": {
            "weather report": {
              "description": "the weather forecast",
              "type": "String"
            }
          },
          "format": "GetWeatherForecast(city, date) -> weather report"
        },
        {
          "name": "RecommendOutfit",
          "Description": "This API recommends an outfit based on the weather conditions.",
          "input_params": {
            "weather": {
              "description": "the weather condition",
              "type": "String"
            }
          },
          "output_params": {
            "outfit details": {
              "description": "the recommended outfit",
              "type": "String"
            }
          },
          "format": "RecommendOutfit(weather) -> outfit details"
        }
    ],

"Query": "What should I wear in London on July 5th?",
"System response":"RecommendOutfit(weather=GetWeatherForecast(city='London', date='July 5th'))"

Given above example, please assume you are a professional assistant who generate multiple reasonable APIs with
their descriptions (not limited to above mentioned ones), User query and system response using at least four APIs in a
nested manner. Let's take a deep breadth and start generating APIs with their descriptions, user query and the
corresponding system response using APIs in a nested manner, please note that the format is the same as the example.
```

## A.2  API FORMAT

```
{
    "name": "PersonName2ID",
```

```
    "Description": "This API is to convert user name to user ID.",
    "input_params": {
        "person_name": {
            "description": "the name of the person",
            "type": "String"
        }
    },
    "output_params": {
        "person_ID": {
            "description": "the ID of the person",
            "type": "Integer"
        }
    },
    "format": "PersonName2ID(person_name) -> person_ID"
}
```

## A.3  PROMPTS FOR BASELINE METHOD

### Zero-Shot Prompt

We have the following functions. Please return function calling according to user instruction with the following format.
APIs: {api info}
user instruction: {user instruction}
please generate the function calling:

### Few-Shot Prompt

We have a list of APIs. Please return function calling according to user instruction.

Here is an example :

APIs:
{"Name": "MakeAppointment", "Description": "This API is to make an appointment.", "input_params": {"hospital_name": {"description": "hospital name", "type": "String"}, "department_name": {"description": "department name", "type": "String"}}, "output_params": {"appointment_status": {"description": "the status of the appointment", "type": "String"}}, "format": "MakeAppointment(hospital_name, department_name) -> appointment status"}
{"Name": "GetDepartment", "Description": "This API is to find the corresponding department given user symptom.", "input_params": {"symptom": {"description": "patient's symptom", "type": "String"}}, "output_params": {"department_name": {"description": "department name", "type": "String"}}, "format": "GetDepartment(symptom) -> department_name"}

user instruction: I'm in zheyi hospital, I have a stomachache and want to make an appointment to see a doctor.
function calling: MakeAppointment (hospital_name='zheyi', department_name=   GetDepartment (symptom = 'stomachache')) "

Given above example, Please generate function calling according to user instruction and the given apis.
APIs: {api info}
user instruction: {user instruction}
please generate the function calling,the format must be the same as example:

## Zero-Shot-CoT Prompt

We have the following functions. Please return function calling according to user instruction with the following format.
APIs: {api info}
user instruction: {user instruction}
please generate the function calling, let's think step by step:

## Few-Shot-CoT Prompt

We have a list of APIs. Please return function calling according to user instruction.

Here is an example :

APIs:
{"Name": "MakeAppointment", "Description": "This API is to make an appointment.", "input_params": {"hospital_name": {"description": "hospital name", "type": "String"}, "department_name": {"description": "department name", "type": "String"}}, "output_params": {"appointment_status": {"description": "the status of the appointment", "type": "String"}}, "format": "MakeAppointment(hospital_name, department_name) -> appointment status"}
{"Name": "GetDepartment", "Description": "This API is to find the corresponding department given user symptom.", "input_params": {"symptom": {"description": "patient's symptom", "type": "String"}}, "output_params": {"department_name": {"description": "department name", "type": "String"}}, "format": "GetDepartment(symptom) -> department_name"}

user instruction: I'm in zheyi hospital, I have a stomachache and want to make an appointment to see a doctor.
thought:
   1. you choose the API named 'GetDepartment', the value for reqiured parameter 'symptom' is 'stomachache', then you will get the output parameter department_name.
   2. then you get hospital_name='zheyi'.
   3. Finally, you choose the API named 'MakeAppointment'.

so the function calling:
MakeAppointment (hospital_name='zheyi', department_name=  GetDepartment (symptom = 'stomachache')) "

Given above example, Please generate function calling according to user instruction and the given apis.
APIs: {api info}
user instruction: {user instruction}
please generate the function calling,the format must be the same as example:

