# OpenReview forum: "Reverse Chain: A Generic Rule for LLMs to Master Multi-API Planning"
_ICLR.cc/2024/Conference — ICLR 2024 Conference Withdrawn Submission_

### Official Review · Reviewer_yWYw · 2023-10-29

**Soundness:** 2 fair
**Presentation:** 2 fair
**Contribution:** 3 good
**Rating:** 3
**Confidence:** 5

**Summary:**

The proposed Reverse Chain can empower large language models (LLMs) with the ability to use external APIs with only prompts. The approach allows LLMs to implement multiple function calling, which is a challenging task due to the complicated relations between different APIs. The authors also conduct experiments that show impressive results, indicating the potential of Reverse Chain in enhancing the performance of LLMs.

**Strengths:**

1. This approach is different from existing approaches that use tool demonstration data to fine-tune LLMs, which can be costly and hard to generalize to new registered APIs. Reverse Chain employs a generic rule to decompose a complicated multi-function calling problem into two types of simple tasks for LLMs, that is API selection and argument completion. This original approach is a significant contribution to the field.

2. The paper presents extensive experiments to evaluate the effectiveness of Reverse Chain. The experiments show that Reverse Chain could efficiently improve the tool-use capabilities of the existing LLMs, e.g., ChatGPT. Moreover, the combination of a generic rule with LLMs enabled Reverse Chain to attain better performance compared to CoT and ReAct.

**Weaknesses:**

1. The method can make the multi-api planning processing more autonomous, but it takes some time to implement the method. Therefore, the complicated process of using the method may be a weakness.

2. The method might take more throughput from the LLM (e.g., the process of selecting API, repeating complete argument). This can cause more computation or more cost on ChatGPT API fees.

3. the paper could benefit from a more detailed comparison with existing approaches. While the paper compares Reverse Chain with CoT and ReAct on ChatGPT, it does not provide a comprehensive experimental comparison on other LLM (e.g., different scales of llama models). Offering a more exhaustive comparison with alternate LLMs would furnish readers with a clearer perspective on the scalability of the Reverse Chain and the efficacy of its selection criteria across different LLMs.

**Questions:**

Please see the weaknesses.

**Details Of Ethics Concerns:**

N.A.

---

### Official Review · Reviewer_62YQ · 2023-10-29

**Soundness:** 3 good
**Presentation:** 3 good
**Contribution:** 3 good
**Rating:** 5
**Confidence:** 3

**Summary:**

This paper presents the ReverseChain, which first selects a final API to handle a given task via LLMs, then ask LLMs to fill the required arguments from user query and avaliable API sets. This process continues until a given task is completed. Extensive numerical experiments indicate an impressive capability of Reverse Chain on implementing multiple function calling.

**Strengths:**

1. Novel idea to use a ReverseChain for API calling, by iteratively select API and fill arguments.

2. Clear paper writting. The reviewer can easily follow the methods and experiments.

3. Strong performance compared to baselines (94.9 vs 88.6)

4. Good ablation and error analysis to show more insights about the methods.

**Weaknesses:**

1. The reviewers are mostly concerning about the datasets for evaluation. The auhors only hand-craft about 500 examples from an existing dataset API-bank, and most of them (about 350) are two-level nested API calling, I am concerning the possiable bias in the dataset. For example, if the APIs have similar functinality, this will be hard the the ReverseChain to select a correct final API. But the authors did not provide more evidence about why this dataset is good for evaluation.

2. The related work is not sufficient. As a very hot topic, there are many studies about LLM-based API calling. The authors only use two short paragraphs in Sention 4. Besides, the reviewer found that the ablation methods in Section 3.2.2 are very similar to the one-step agent and sequential agent proposed in TPTU [1], the authors could discuss their differences if possiable. [1] TPTU: Task Planning and Tool Usage of Large Language Model-based AI Agents.


3. small writing issues. e.g., "they may contain thoughts,action" ==> thoughts, actions; "T represents the temperature of GPT-3.5."  ==> remove it.

**Questions:**

1. Could the authors provide more evidence about why the hand-crafted dataset is good for evaluation?

2. Could the authors add more related works and disscuss the differences between the ablation methods in Section 3.2.2 and the one-step agent and sequential agent proposed in TPTU?

3. If the APIs have similar functinality, this will be hard the the ReverseChain to select a correct final API. Can ReverseChain works well in this setting?

---

> ### Author Response · Authors · 2023-11-21
> **Difference from TPTU**
>
> Thank you for your suggestions.
> Q: The related work is not sufficient.
> A: We will add more related work and comparison with our work in the next version of the paper.
>
> Q: The reviewer found that the ablation methods in Section 3.2.2 are very similar to the one-step agent and sequential agent proposed in TPTU [1], the authors could discuss their differences if possible.
>
> A: There is a big difference between our work Reverse Chain and one-step agent in TPTU.
> TPTU is executed in forward order when generating a plan, while Reverse Chain is executed in reverse order when generating a plan. Starting from the last API, it infers the APIs required in the previous step in reverse.
>
> Take the example in the TPTU paper as an example:
> Question: How much budget is required to provide a 100$ incentive for each colleague who has worked for five years?
>
> In TPTU, the planning is:
> 1. SQL generator: “Figuring out how many colleague who has worked for five years from the database; taking it as X.”
> 2. Python generator: “Calculating the value of 100*X with a calculator”
>
> However, in Reverse Chain, First, it is recognized that the final intent is calculation, so calculator is chosen. The calculator parameter is missing X, and then the SQL generator is chosen to generate the number of colleagues who has worked for five years from the database, which is X.

---

> > ### Comment · Reviewer_62YQ · 2023-11-22
> > **Possiable bias in the datasets for evaluation?**
> >
> > Thanks for the clarification. It helps for understanding the difference between Reverse Chain and TPTU.
> >
> > I am also concerning about the datasets for evaluation. Could the authors give some explaination? I would be happy to increase the score if this main concern is addressed.

---

### Official Review · Reviewer_xS6A · 2023-11-03

**Soundness:** 2 fair
**Presentation:** 2 fair
**Contribution:** 2 fair
**Rating:** 3
**Confidence:** 3

**Summary:**

In this paper, the authors present an in-context learning approach to enable LLMs use API calls to complete user requests. The authors present a general approach called "Reverse Chain", which can also extend to serving user requests which involve multiple levels of API calls to fulfill top-level API request. The authors compare with different zero-shot and few-shot approaches and show interesting performance of their approach.

**Strengths:**

- I think the paper is well-motivated. Having the LLMs be able to output API calls is useful for lots of applications to enable the model access a lot of different information that it doesn't have knowledge of directly, or is too big to fit in context.

- The biggest strength of this approach is its simplicity. The approach is intuitive and the ablations considered by the authors help understand the approach better.

**Weaknesses:**

- I am mainly concerned about the novelty of this approach. The authors do mention the relevant recent papers in their related work section, however, I feel that the authors haven't done an adequate job of comparing their work with existing work. (more on this in questions section)

- I think the experiments can be expanded a bit more. The current dataset seems limited and small.

**Questions:**

- How does this paper compare more directly with the depth-first approach presented in ToolLLM paper [1]? I understand that ToolLLM paper didn't directly look at composition multi-tool problem, but shouldn't it be straightforward to extend their DFS approach to do this? And won't that be similar to what you're proposing?

- How does this method scale with # of APIs? For instance, the ToolLLM[1] paper had >16,000 APIs in their dataset. This would require some shortlisting using a retriever.

- I think the authors should also include # of API calls in their experimental results so that readers can understand the trade-off between cost/latency and the quality of the final outputs. My guess is that for applications that need very few APIs and where the API names are very distinct, simple zero-shot methods would suffice.

- It'll be useful to break down the analysis based on the length of API call or num. of arguments in the API call and the performance of Reverse Chain and the corresponding baselines.

- Summarizing the above two steps into a discussion section to directly point out under what circumstances Reverse Chain would be useful would be greatly beneficial for researchers and practitioners alike.


References

[1] Qin, Y., Liang, S., Ye, Y., Zhu, K., Yan, L., Lu, Y., ... & Sun, M. (2023). Toolllm: Facilitating large language models to master 16000+ real-world apis. arXiv preprint arXiv:2307.16789.

---

### Official Review · Reviewer_2PHd · 2023-11-06

**Soundness:** 3 good
**Presentation:** 3 good
**Contribution:** 3 good
**Rating:** 5
**Confidence:** 4

**Summary:**

The paper presents an approach called Reverse Chain for compositional API calling. The approach uses prompts to start off with the Goal API to be determined followed by filling in the arguments of the APIs iteratively either by extracting from the user text or calling another API. The approach is fairly straight forward and the experimental results shows that it works very well. The paper is also well written and understandable. Backward chaining in reasoning techniques has been effectively explored for LLMs in this work. The ablation study and the error analysis evidences further on the reverse chain approach.

Primary concerns of this work are as follows:
1. Generalizability of the approach: This approach is specific to single goal —> compositional API calling work. When there are aspects of multiple goals in a specific user query it is unclear how this approach can be adapted. Furthermore, this does set an iterative pipeline of two tasks API selection and Argument Completion in a way that it gets hard for it to be more generalizable in the context of plug-ins and other API tasks. This does beg the question “Would APIs be only used in such scenarios?”
2. Dataset for evaluation: This brings a larger concern — looking at the prompt and the few shot examples, it’s very clear that the dataset is constructed exactly for the reverse chain to work. The results for level 1 and level 2 clearly shows significant performance where the API/Tool calling abilities of large language models is yet to be improved. This is also true for few shot cot which is doing much better than ReAct.
    1. Furthermore, given 126 APIs in the dataset — does each instance during test have all the 126 APIs? How many domains does these APIs belong to?
    2. 492 samples — What’s the distribution of the 126 APIs in the 492 samples.
    3. More details of the dataset is necessary to interpret the results better.

**Strengths:**

1. Paper has a very simple but effective approach to solve compositional API calling.
2. Results show the evidence and the paper is presented well

**Weaknesses:**

1. Questions about Generalizability of the approach
2. Dataset details are not self contained in the paper which is necessary to interpret the results

**Questions:**

In the Summary

---

### Official Review · Reviewer_75Tz · 2023-11-10

**Soundness:** 2 fair
**Presentation:** 2 fair
**Contribution:** 2 fair
**Rating:** 5
**Confidence:** 4

**Summary:**

This article proposes a Reverse Chain to empower LLMs with capabilities to use external APIs with only prompts. In this reverse chain, the LLMs are asked to fill in the required arguments based on the query/context before checking with the user. A good attempt on in-contextual learning methods comparison but lacked research contribution.

**Strengths:**

Reverse Chain concept for multi api planning
Good contribution on tool comparison & in-context method comparison

**Weaknesses:**

Some of the weakness of this article are as follows:
1. The concept of reverse chain can be applied to simple tasks for ex. API selection and argument completion.
2. How reverse chain proved to be effective in multi-API planning is not delineated in detail
3. This article focuses on comparing multiple tools or methods in API calling & planning & reduces research contribution.
4. Experimental results too discusses on comparing different methods of in-context learning against Reverse Chain but needed empirical evaluation of the proposed method to be accepted in reputed ICLR.

**Questions:**

The complexity on the API nesting needs to be at a higher in order to prove Reverse Chain out passes the other in-context learning approaches? Why the nesting of APIs are set at 2, 3 and 4 levels? Is there any experiments conducted with larger nesting capabilities?
Final date set has about 126 APIs and 492 samples; what about the number of queries that was tested against the metrics?
The authors have made a statement "This framework enables the LLM to generate responses by following a logical sequence of thinking, taking action, and observing the results." How this can be achieved needs to be explored in detail with Reverse Chain?
In section 3.3 -> It summarizes the numbers based on wrong final tool/ wrong argument api and value but does not state any where against how many number of times or queries used against?